# Impact of Coated Zinc Oxide Nanoparticles on Photosystem II of Tomato Plants

**DOI:** 10.3390/ma16175846

**Published:** 2023-08-26

**Authors:** Panagiota Tryfon, Ilektra Sperdouli, Ioannis-Dimosthenis S. Adamakis, Stefanos Mourdikoudis, Michael Moustakas, Catherine Dendrinou-Samara

**Affiliations:** 1Laboratory of Inorganic Chemistry, Department of Chemistry, Aristotle University of Thessaloniki, 54124 Thessaloniki, Greece; tryfon.giota@gmail.com; 2Institute of Plant Breeding and Genetic Resources, Hellenic Agricultural Organization-Dimitra, 57001 Thessaloniki, Greece; ilektras@bio.auth.gr; 3Section of Botany, Department of Biology, National and Kapodistrian University of Athens, 15784 Athens, Greece; iadamaki@biol.uoa.gr; 4Biophysics Group, Department of Physics and Astronomy, University College London, London WC1E 6BT, UK; s.mourdikoudis@ucl.ac.uk; 5UCL Healthcare Biomagnetics and Nanomaterials Laboratories, 21 Albemarle Street, London W1S 4BS, UK; 6Department of Botany, Aristotle University of Thessaloniki, 54124 Thessaloniki, Greece

**Keywords:** nanoagrochemicals, inorganic nanoparticles, chlorophyll fluorescence, photoinhibition, photoprotection, electron transport rate, reactive oxygen species, effective quantum yield of PSII photochemistry (Φ*_PSII_*), maximum photochemistry (F*v*/F*m*)

## Abstract

Zinc oxide nanoparticles (ZnO NPs) have emerged as a prominent tool in agriculture. Since photosynthetic function is a significant measurement of phytotoxicity and an assessment tool prior to large-scale agricultural applications, the impact of engineered irregular-shaped ZnO NPs coated with oleylamine (ZnO@OAm NPs) were tested. The ZnO@OAm NPs (crystalline size 19 nm) were solvothermally prepared in the sole presence of oleylamine (OAm) and evaluated on tomato (*Lycopersicon esculentum* Mill.) photosystem II (PSII) photochemistry. Foliar-sprayed 15 mg L^−1^ ZnO@OAm NPs on tomato leaflets increased chlorophyll content that initiated a higher amount of light energy capture, which resulted in about a 20% increased electron transport rate (ETR) and a quantum yield of PSII photochemistry (Φ*_PSII_*) at the growth light (GL, 600 μmol photons m^−2^ s^−1^). However, the ZnO@OAm NPs caused a malfunction in the oxygen-evolving complex (OEC) of PSII, which resulted in photoinhibition and increased ROS accumulation. The ROS accumulation was due to the decreased photoprotective mechanism of non-photochemical quenching (NPQ) and to the donor-side photoinhibition. Despite ROS accumulation, ZnO@OAm NPs decreased the excess excitation energy of the PSII, indicating improved PSII efficiency. Therefore, synthesized ZnO@OAm NPs can potentially be used as photosynthetic biostimulants for enhancing crop yields after being tested on other plant species.

## 1. Introduction

Photosynthesis, a vital process in the natural world, converts light energy into chemical energy, sustaining life on earth. Central to this process is photosystem II (PSII), a protein complex responsible for capturing and utilizing light energy to generate electron flow and drive the synthesis of energy-rich molecules. The efficient functioning of PSII is essential for optimal photosynthetic performance and biomass production in plants. Impairments in PSII functionality can lead to a decline in photosynthetic capacity, hampering plant growth and development, and ultimately reducing crop productivity [1,2]. Therefore, the assessment of PSII functionality by the method of chlorophyll fluorescence analysis, which is a widely recognized technique in plant physiology, provides valuable insights into the responses of plants to environmental stimuli [3,4,5]. In particular, this technique has been demonstrated to be successful in evaluating the impact of stress on photosynthetic traits [6,7] and has been documented as the most appropriate method to detect the toxicity caused by nanoparticles (NPs) on plants [6,7,8,9,10].

In recent years, the development of photocatalytic reaction systems has garnered considerable attention as a promising solution to tackle the pressing challenges related to energy and the environment [11,12,13]. One particular area of interest involves the synthesis of an ideal semiconductor with more electron–hole pairs and a large surface area capable of preventing the recombination of the electron carriers [14]. In this venue, engineered inorganic nanoparticles (EINPs) meet phytonanotechnology [15]. EINPs with photoactive properties are candidates for enhancing photosynthetic processes in plants. Zinc is an essential trace element in plants, playing a crucial role in various physiological and metabolic activities [16]. It interferes as a cofactor in the enzymes involved in respiration, photosynthesis, and hormone synthesis, contributing to plant development and overall metabolic functions [17,18]. Thus, zinc oxide nanoparticles (ZnO NPs) have gained significant attention due to their unique properties, including their high electron mobility, wide bandgap, and exceptional photocatalytic activity [19,20]. They are also widely used in several medical-based applications, such as in cosmetics, as drug delivery agents and for their antibacterial activities against both Gram-positive and Gram-negative pathogenic strains [21].

The impact of ZnO NPs depends on the plant species, the time of exposure, the concentration used and their size and morphology [22,23]. Recent investigations have revealed the positive effects of foliar-applied ZnO NPs on growth, physiology, and the photosynthetic rate of coffee plants [24]. Foliar treatment of ZnO NPs at 1000, and 2000 mg L^−1^, efficiently improved the physiological responses of pepper plants, including their leaf fluorescence parameters and H_2_O_2_ levels under salinity conditions [25]. Additionally, experimental results revealed that ZnO NPs reduced drought and heat stress on wheat plants by improving biochemical metabolism [26].

Polyol-coated CuZn bimetallic NPs and ZnO nanoflowers were assessed for their potential as antifungal agents and nano-agrochemicals against fungal species, and also for their enhancement effects on photosynthesis [9,27,28]. Despite the potential benefits of ZnO NPs, there is still a lack of comprehensive understanding regarding their underlying reaction mechanisms, which has restricted the extended use of ZnO-coated NPs. Therefore, conducting fundamental research in this field is important to enhance the overall efficiency of these photocatalytic systems.

Herein, the intrinsic properties of ZnO NPs, such as the electron conductivity in the proportion of oleylamine (OAm) used as a capping agent, were explored on the PSII and various photosynthetic parameters in tomato plants. The ZnO@OAm NPs were solvothermally prepared in the sole presence of OAm in a facile and reproducible procedure. The use of OAm as a capping agent provides stable and well-dispersed hydrophobic and/or partly hydrophilic NPs based on the method of synthesis [29]. Meanwhile, amines are essential for growth and development and their metabolism appears to be coordinated with the cell cycle. Interestingly, recent findings suggest that the absorption efficacy of hydrophobic particles in tomato plant root penetration surpasses that of hydrophilic ones by roughly 1.3 times [30]. In this study, we undertake the foliar application of hydrophobic ZnO@OAm NPs that are expected to improve photosynthetic efficiency more effectively than root application. Taking into account that photosynthetic function is a significant measurement of phytotoxicity and an assessment tool prior to a large-scale agricultural application, the as-prepared NPs were applied on tomato leaflets (*Lycopersicon esculentum* Mill.) using a foliar spray while chlorophyll content, reactive oxygen species (ROS) generation and photosynthetic function were evaluated.

## 2. Materials and Methods

All chemicals and reagents for ZnO@OAm NPs synthesis were of analytical grade and were used without any further purification: Zinc (II) acetylacetonate hydrate [Sigma-Aldrich, St. Louis, MO, USA, *M* = 263.61 g mol^−1^, Zn(acac)_2_], Oleylamine [Merck, Darmstadt, Germany, *M* = 267.493 g mol^−1^, OAm], and ethyl alcohol (*M* = 46.07 g mol^−1^).

### 2.1. Solvothermal Synthesis of ZnO@OAm NPs

The solvothermal approach is a versatile technique for crafting a diverse range of materials, from metals and ceramics to semiconductors and polymers. This procedure employs solvents under varying pressures and temperatures, promoting precursor interactions offering good homogeneity in size and morphology. Meanwhile, the adaptability, cost-effectiveness and simplicity of this synthetic approach are its core strengths [31].

The synthesis of ZnO@OAm NPs was performed according to a previous study [32] with certain modifications. Briefly, Zn(acac)_2_ (1.06 mmol) was dissolved in 4 mL of OAm, and mixed well under stirring at 30 °C for 15 min. The resulting solution was transferred into a Teflon-lined stainless-steel autoclave to set out a solvothermal process. The reaction was conducted at 200 °C with a ramp time of 4 °C min^−1^, and a hold time of 8 h, followed by natural cooling to room temperature. Afterward, the synthetic mixture was centrifuged at 5000 rpm for 20 min and washed three times with ethanol. The supernatants were discarded each time for the removal of byproducts and the unreacted precursor.

### 2.2. Physicochemical Characterization of Nanoparticles

The crystal structure and crystallinity of NPs was investigated through X-ray diffraction (XRD) performed on a Philips PW 1820 diffractometer in the 2θ range from 10 to 90°, with monochromatized Cu Kα radiation (λ = 1.5406 Å). The average crystalline grain size was calculated based on Debye–Scherrer’s equation [33]. A Nicolet series FT-IR spectrometer (Nicolet iS20, Thermo Fisher Scientific, Waltham, MA, USA) with a monolithic diamond ATR crystal was used to acquire the FT-IR spectrum (4000−450 cm^−1^) of NPs. The average particle size and the morphology were determined by transmission electron microscopy (TEM) with a JEOL JEM 1200−EX (Tokyo, Japan) electron microscope at an acceleration voltage of 120 kV. Thermogravimetric analysis (TGA) was used to determine the thermal stability and the amount of organic coating of NPs using a SETA−RAM SetSys-1200 at a heating rate from 30 °C to 800 °C (10 °C min^−1^) under an N_2_ atmosphere. The optical properties of NPs in an ethanol/water (1:3) solution and their bandgap value were identified (15 mg L^−1^) using a UV-Vis spectrophotometer (V-750, Jasco, Tokyo, Japan) and based on Tauc’s formula [34,35]. Dynamic light scattering (DLS) analysis was used to define the size-distribution profile of ZnO@OAm NPs (15 mg/L), polydispersity index (PDI), and the ζ-potential (mV) measurements to assess the surface charge of the particles using a Zetasizer (Nano ZS Malvern apparatus VASCO Flex™ Particle Size Analyzer NanoQ V2.5.4.0, Malvern, UK) at 25 °C. The measurements were performed in ethanol/water mixture (1:3) with addition of HCl (0.1 M) and sonication for 15 min.

### 2.3. Plant Material and Growth Conditions

Tomato (*Lycopersicon esculentum* Mill. cv. Galli) plants were obtained from the market and then transferred to a greenhouse with a photosynthetic photon flux density (PPFD) of 600 ± 10 μmol quanta m^−2^ s^−1^ and a 14 h photoperiod, 25 ± 1/20 ± 1 °C day/night temperature and relative humidity 65 ± 5/75 ± 5% day/night.

### 2.4. Exposure of Tomato Plants to ZnO@OAm NPs

Tomato plants were sprayed either with distilled water (control) or with 15 mg L^−1^ ZnO@OAm NPs. All treatments were performed with 3 to 5 plants and three independent biological replicates.

### 2.5. Chlorophyll Content Measurements

The chlorophyll content was measured photometrically with a portable Chlorophyll Content Meter (Model Cl-01, Hansatech Instruments Ltd., Norfolk, UK), using dual-wavelength optical absorbance (660 nm and 940 nm wavelength). Values were expressed in relative units [36,37]. The measurements were carried out on tomato leaflets 90 min after being sprayed with distilled water (control) or with 15 mg L^−1^ ZnO@OAm NPs.

### 2.6. Chlorophyll Fluorescence Analysis

Chlorophyll fluorescence measurements were performed with an Imaging PAM Fluorometer M-Series MINI-Version (Heinz Walz GmbH, Effeltrich, Germany) in dark-adapted (20 min) tomato leaflets as described in detail previously [38]. The measurements were carried out on tomato leaflets 30 and 90 min after the tomato plants were sprayed with distilled water (control) or with 15 mg L^−1^ ZnO@OAm NPs. In each tomato leaflet, 10 areas of interest (AOI) were selected and chlorophyll fluorescence measurements were conducted with two actinic light (AL) intensities, at 600 μmol photons m^−2^ s^−1^, corresponding to the growth light (GL) intensity of plants, or at 1000 μmol photons m^−2^ s^−1^, corresponding to a high light (HL) intensity, following the protocols described by Moustaka et al. [38] and Oxborough and Baker [39]. Chlorophyll fluorescence parameters, estimated by Imaging Win V2.41a software (Heinz Walz GmbH, Effeltrich, Germany), are described in Appendix A.

### 2.7. Imaging of Hydrogen Peroxide (H_2_O_2_) Generation

The imaging of hydrogen peroxide (H_2_O_2_) generation in the tomato leaflets was performed 30 min after tomato plants were sprayed with distilled water (control) or with 15 mg L^−1^ ZnO@OAm NPs as described previously [40]. Leaves were incubated in the dark for 30 min with 25 μM 2′,7′-dichlorofluorescein diacetate (DCF-DA, Sigma Aldrich, Chemie GmbH, Schnelldorf, Germany) [40] and observed with Zeiss AxioImager Z2 epi-fluorescence microscope equipped with an AxioCam MRc5 digital camera [41].

### 2.8. Statistical Analysis

Three independent experiments with at least three biological replicates were conducted and the results are presented as mean ± SD. Significant differences were determined using ANOVA, followed by Tukey’s post hoc tests for each parameter. The variance homogeneity test was used to verify the parametric distribution of data. Values were considered to be significantly different at *p <* 0.05.

## 3. Results and Discussion

### 3.1. Physicochemical Characterization of ZnO@OAm NPs

The purity and crystallite size of the prepared ZnO@OAm NPs were measured through XRD analysis (Figure 1). XRD revealed that the ZnO@OAm NPs had a hexagonal wurtzite ZnO structure (space group P63mc (186), JCPDS card #89-0510), which is consistent with previous studies [32,42,43]. The main diffraction peaks corresponding to the (100), (002), and (101) crystal planes are observed, indicating the high phase purity of the NPs. Moreover, diffraction peaks of high intensities were present at 2θ = 18–25° and attributed to the well-crystallized OAm on the NP surface. The crystallite size of the NPs was estimated to be 19 nm using the Scherrer formula, based on the (101) reflection plane. A crystallinity of approximately 91% was found for the ZnO@OAm NPs, a value that appears to be influenced by the solvothermal approach and/or the ethanol that was employed as a solvent during the repeating washing steps [44]. The lattice parameters of the ZnO@OAm NPs were calculated to be *α* = *b* = 3.2635 Å and *c* = 5.2349 Å. The d-spacing values for the three major diffraction peaks were found to be 2.82, 2.61, and 2.49 Å, respectively.

The morphological and structural characteristics of the ZnO@OAm NPs were explored by TEM imaging. The TEM images revealed that the particles had irregular shapes (Figure 2a,b) with a narrow size distribution attributed to the OAm that provides a good control over their dimensions. The size distribution histogram (Figure 2b; insert) showed a Gaussian curve-fitting, indicating that the particles were well dispersed with an average diameter of about 9 ± 1.25 nm. This value is close to the corresponding values from the XRD pattern (Figure 1), indicating their single-crystal character.

FT-IR analysis was used to identify the functional groups and chemical bonds of the ZnO@OAm NPs. The FT-IR spectrum (Appendix A) revealed the presence of characteristic peaks corresponding to the metal–oxygen bond (Zn–O) at 451 cm^−1^ and various organic functional groups. The bands at 2915 and 2842 cm^−1^ corresponded to *ν_as_*(C–H) and *ν_s_*(C–H) stretching vibrations, respectively, while the bands at 3311 and 3240 cm^−1^ were attributed to the N–H stretching bond of the primary and secondary amino groups. Furthermore, peaks of low intensity at 1573 cm^−1^ (ΝH_2_), 1462 cm^−1^ *δ*(CH_3_), 711 cm^−1^ *δ*(–C–C–) were observed [45], and a weak peak at 1646 cm^−1^ was linked to the stretching vibration of the double bond *δ*(–C=C) [46].

Thermal stability and organic coating content (% *w*/*w*) of ZnO@OAm NPs were assessed by TGA up to 800 °C (Appendix A). The decomposition of the OAm occurred in two unequal regions of mass reduction at 160–300 °C (26% *w*/*w*) and 400–450 °C (4% *w*/*w*); in the first step, hydrocarbon chains were removed while the nitrogen head groups were burnt off in the second step. The nitrogen head groups coordinated on the surface of the NPs exhibited thermal stability, as they persisted on the surface and underwent decomposition at higher temperatures. The behavior of the OAm double layer was similar to that reported previously for oleylamine-capped NPs [32,47]. A total weight loss of 30% *w*/*w* was determined for the NPs’. The thermal trend of the sample is influenced by its structure, homogeneity, and composition [48]. A relatively small particle size, such as that of the 19 nm ZnO@OAm NPs, results in a large surface area that leads to the release of decomposition products at a faster rate during heating, as approved.

The as-prepared ZnO@OAm NPs were considered partly hydrophilic as they were dispersed in an ethanol/water mixture (1:3). The UV-Vis absorption spectrum of the ZnO@OAm NPs (Appendix A) exhibited a characteristic absorption peak at 375 nm, which is typical of the absorption of the ZnO semiconductor [49,50,51]. Additionally, the absorbance at 226 nm is attributed to the crystallized OAm, indicating its stability on the NPs in the solution, as was also confirmed in solid form through the TGA and XRD analyses (Figure 1 and Appendix A). The bandgap value of ZnO@OAm NPs was estimated at 3.14 eV, which aligns well with the absorption properties of PSII in plants. The close match between the bandgap of ZnO NPs and the absorption spectrum of PSII allows the NPs to serve as photosensitizers, absorbing light energy and transferring it to the photosystem pigments in plants.

The DLS technique was employed to determine the hydrodynamic size of ZnO@OAm NPs in the ethanol/water mixture. The size distribution of the particles presented a stable colloidal suspension of NPs with an average size of 73 nm (Appendix A), much smaller than the naked commercial ZnO NPs (140.2 ± 2.7 nm) [52]. This size corresponds to the agglomeration of the NPs, a common phenomenon due to the presence of ions and the organic coating attached to the surfaces of the ZnO NPs in the suspension [53]. Notably, monodispersed size distribution and homogeneity were observed with a PDI value at 0.018 attributed to the OAm, that is known to improve colloidal stability [54,55]. The ζ-potential of the NPs was measured at −14.71 ± 0.38 mV (Appendix A), ascertaining the efficacy of OAm in the stabilization of the particles.

### 3.2. Impact of ZnO@OAm NPs on PSII Photochemistry

#### 3.2.1. Maximum Efficiency of PSII and Efficiency of the Oxygen-Evolving Complex

The relative chlorophyll content of tomato leaflets 90 min after being sprayed with 15 mg L^−1^ ZnO@OAm NPs increased by 12% compared to tomato leaflets that were sprayed with distilled water (control) (Figure 3a). In contrast, the maximum efficiency of PSII photochemistry (F*v*/F*m*) of the tomato leaflet, decreased significantly at 30 and 90 min after being sprayed with ZnO@OAm NPs, compared to that of the controls (Figure 3b). The same trend was noticed on the efficiency of the oxygen-evolving complex (OEC) (F*v*/F*o*), whose functionality decreased (*p* < 0.05) by 6% and 7% at 30 min and 90 min after being sprayed with ZnO@OAm NPs, NPs, compared to that of the controls (Figure 3c). A decreased PSII maximum efficiency (F*v*/F*m*), compared to the control (Figure 3b), suggests a degree of photoinhibition [56,57,58,59]. However, despite PSII photoinhibition, the light- harvesting complexes (LHCII) continue to absorb light [60]. ZnO NPs have been shown to increase chlorophyll content in a concentration-dependent way [61,62]. This increase in chlorophyll content can be justified by the involvement of zinc in protochlorophyllide and chlorophyll synthesis [62,63]. Increased chlorophyll content is accompanied by larger light-harvesting complexes (LHCs) and also by a higher amount of light energy capture [64,65].

Malfunction of the OEC results in donor-side photoinhibition [66,67,68]. It seems that the ZnO@OAm NPs interfered with the oxygen-evolving Mn_4_CaO_5_ cluster of PSII, causing the malfunction that was reflected in the maximum photochemistry (F*v*/F*m*), that decreased. Photosynthetic water oxidation by PSII is an appealing process because it sustains life on earth [69].

When there is a malfunction of the OEC, it will not efficiently reduce the chlorophyll molecule of the PSII reaction center (P680+), resulting in harmful oxidations in PSII [70]. Thus, the donor side photoinhibition is frequently connected to ROS creation, e.g., hydrogen peroxide (H_2_O_2_), which can be oxidized to the superoxide radical (O_2_^•−^) or reduced to the hydroxyl radical (HO^•^) [70].

#### 3.2.2. Absorbed Light Energy Partitioning and Non-Photochemical Quenching

The light energy absorbed by the LHCII partitions PSII photochemistry (Φ*_PSII_*), regulating non-photochemical energy loss in PSII (Φ*_NPQ_*), and nonregulated energy loss in PSII (Φ*_NO_*), with the sum of all being equal to 1 [71]. The effective quantum yield of PSII photochemistry (Φ*_PSII_*) (Figure 4a) increased (*p* < 0.05) by 21% and 19% at 30 and 90 min after being sprayed with ZnO@OAm NPs at growth light (GL, 600 μmol photons m^−2^ s^−1^), and by 26% and 27% at high light (HL, 1000 μmol photons m^−2^ s^−1^), respectively, compared to the controls.

Elshoky et al. [72] reported that foliar spray with 200 mg L^−1^ ZnO NPs did not influence the photosystem functions of *Pisum sativum*, while 400 mg L^−1^ ZnO-Si NPs had positive consequences on Φ*_PSII_* and PSI photochemistry. As a result of the increase in Φ*_PSII_*, the regulated non-photochemical energy loss in PSII (Φ*_NPQ_*) decreased (*p* < 0.05) under GL at 30 and 90 min after being sprayed with ZnO@OAm NPs (−16% and –−13%, respectively) and also by (*p* < 0.05) under HL (−13% and –−11%, respectively), compared to that of the controls (Figure 4b). The nonregulated energy loss in PSII (Φ*_NO_*) remained the same as that of the control under GL but increased under HL i (*p* < 0.05) (6%) compared to the control at 30 min after being sprayed with ZnO@OAm NPs (Figure 4c). However, 90 min after being sprayed with ZnO@OAm, the NPs returned to that of the control level (Figure 4c). Φ*_NO_*, which is a measure of the singlet-excited oxygen (^1^O_2_) generation [64,73,74], increased under HL at 30 min after being sprayed (Figure 4c), suggesting an increase in ROS production.

Singlet-excited oxygen (^1^O_2_) is created from the triplet-excited state of chlorophyll (^3^Chl*) which is generated through an intersystem crossing from the singlet-excited state of chlorophyll (^1^Chl*) [3,75,76]. The increased ^1^O_2_ formation motivates subsequent photoinhibition [76,77,78,79]. Chlorophyll molecules play a pivotal role as the primary pigments responsible for capturing light energy and channeling it to the reaction centers, with consequent electron transport [75,80,81,82]. The increased chlorophyll content with ZnO@OAm NPs resulted in increased ^1^O_2_ formation.

The dissipation of excitation energy as heat by NPQ decreased (*p* < 0.05) under both GL (−18% and −12% at 30 and 90 min after being sprayed with ZnO@OAm NPs, respectively) and HL (−17% and −12% at 30 and 90 min after being sprayed with ZnO@OAm NPs, respectively), compared to that of the controls (Figure 4d). The increased ^1^O_2_ formation with ZnO@OAm NPs under HL, 30 min after being sprayed with ZnO@OAm NPs (Figure 4c) could be due to the decreased NPQ (Figure 4d). NPQ is the photoprotective mechanism that prevents ROS formation [83,84,85,86,87,88]. Enhanced dissipation as heat by nonphotochemical quenching (NPQ) decreases photoinhibition [89,90]. Non-photoinhibited control leaves showed enhanced NPQ (Figure 4d).

#### 3.2.3. Photochemical Quenching and Efficiency of Open PSII Centers

The fraction of open PSII reaction centers (q*p*) (Figure 5a), that is the redox state of quinone A (QA), increased (*p* < 0.05) under GL (18%, at 30 and 90 min after being sprayed with ZnO@OAm NPs) and HL (22% and 25% at 30 and 90 min after being sprayed with ZnO@OAm NPs, respectively), compared to that of the controls (Figure 5a). The efficiency of the open PSII reaction centers 30 min after being sprayed with ZnO@OAm NPs, increased (*p* < 0.05) (3%) under both GL and HL, compared to that of the controls (Figure 5b). However, under both GL and HL, it did not differ to that of the controls at 90 min after being sprayed with ZnO@OAm NPs (Figure 5b).

An increased Φ*_PSII_* can be attributed either to the fraction of open PSII reaction centers (q*p*) or to the efficiency of these centers (F*v*’/F*m*’) [91]. The increased ΦPSII at 30 min after being sprayed with ZnO@OAm NPs was due to the increases in both q*p* (Figure 5a) and F*v*’/F*m*’ (Figure 5b), while after 90 min, it was due only to the increase in q*p* (Figure 5a).

#### 3.2.4. Electron Transport Rate and Excess Excitation Energy

The electron transport rate (ETR) increased (*p* < 0.05) at 30 and 90 min after being sprayed with ZnO@OAm NPs, compared to that of the controls, under both GL (21% and 19%, respectively) and HL (26% and 27%, respectively) (Figure 5c). On the contrary the excess excitation energy (EXC) decreased (*p* < 0.05) at 30 and 90 min after being sprayed with ZnO@OAm NPs, compared to that of the controls under both GL (−18% and −20%, respectively) and HL (−14% and −18%, respectively) (Figure 5d). The increased ETR of the tomato plants after the foliar spray with ZnO@OAm NPs was due to a decreased NPQ [92,93]. Photoinhibition decreased NPQ, enhancing the electron transport rate [60].

Chlorophyll molecules are the main pigments that absorb the light quanta and transfer the energy to the reaction centers. An increased chlorophyll content results in larger light-harvesting complexes (LHCs) and leads to a higher amount of light energy capture, causing an increased PSII quantum yield (Φ*_PSII_*) [64,65]. An improved photosynthetic function after exposure to ZnO NPs has been attributed to the enhancement in light acquisition that led to enhanced photosynthesis and also to shielding the chloroplast from aging [62,94]. Tomato plants, after being sprayed with ZnO@OAm NPs, possessed increased Φ*_PSII_* (Figure 4a) and ETR values (Figure 5c), but also decreased excess excitation energy at PSII (Figure 5d), indicating improved PSII efficiency. Increasing the photosynthesis of food crops in order to undertake the huge demand for food on earth is a true challenge for crop breeders and plant scientists [95,96,97]. Thus, the challenge of improving crop performance by enhancing the photosynthetic efficiency of crop plants is a crucial and high-significance research issue [98,99]. The goal of enhancing photosynthetic efficiency can be achieved via a better allocation of the absorbed light energy [100]. EINPs can enhance crop yield by stimulating photosynthesis and inhibiting various pathogens as a result of their antimicrobial properties [9].

### 3.3. Impact of ZnO@OAm NPs on Reactive Oxygen Species

The slightly increased H_2_O_2_ generation on the minor lamina veins and the leaf midrib (Figure 6) is in accordance with the enhanced ROS accumulation also observed in the same regions of *Brassica juncea* treated with ZnO NPs [62]. H_2_O_2_ generation (Figure 6) and ^1^O_2_ generation (under HL) (Figure 4c) both increased on tomato leaflets 30 min after being sprayed with 15 mg L^−1^ ZnO@OAm NPs, suggesting an increased ROS level.

The slight increase in H_2_O_2_ generation that we observed, owing to the donor-side photoinhibition, is due to a malfunction of the OEC [70]. The NPQ mechanism, by dissipating excess light energy, protects the photosynthetic apparatus from the damaging effects of ROS [75,76,80,101]. A small amount of ROS is needed to maintain life, while a slightly increased ROS level activates molecular tolerance mechanisms, which are considered to be positive. However, an elevated level of ROS is deemed to be damaging to plants [59,76,102,103,104,105,106]. The ability of ZnO NPs to display anticancer and antibacterial activities is attributed to their ability to provoke ROS generation [107,108,109]. However, ZnO NPs have been characterized as safe by the US Food and Drug Administration (FDA) and are permitted for use as effective drug delivery systems [109,110].

## 4. Conclusions

Several engineered inorganic-based NPs can serve as photosensitizers, absorbing light energy and transferring it to the photosystem pigments in plants. However, ZnO NPs possess a direct broad bandgap of approximately 3.3 eV, aligning with the near-UV spectrum. Their high exciton binding energy at ambient conditions endows them with distinct photophysical attributes. Nevertheless, differentiation in size, shape, structure, and surface properties affect their performance in correlation with the plant species. In the present study, irregular-shaped ZnO NPs coated with 30% oleylamine were synthesized via solvothermal process that can be considered as a forerunner to large-scale production. The sole presence of OAm influenced the shape, the percentage, and the layering mode of the coating, resulting in partly hydrophilic ZnO NPs. Thirty minutes after the foliar application of 15 mg L^−1^ ZnO@OAm NPs on tomato plants, chlorophyll content, hydrogen peroxide (H_2_O_2_), and singlet-excited oxygen (^1^O_2_) generation increased. However, despite the increased ROS accumulation, an enhanced PSII quantum yield (Φ*_PSII_*) was recorded due to the increased chlorophyll content that resulted in larger light-harvesting complexes (LHCs) and a higher amount of light energy capture. The improved photosynthetic function after exposure to ZnO@OAm NPs can be attributed to the enhancement in light acquisition that led to enhanced photosynthesis. Therefore, oleylamine-coated ZnO NPs can potentially be used as photosynthetic biostimulants for enhancing crop yields after being tested on other plant species. However, further research is required to examine the impact of different coatings of ZnO NPs on ROS generation and photosynthetic function, as well as the possible differential mechanisms of action of various percentages of oleylamine as a capping agent on photosynthetic function, for the extensive application of ZnO NPs as a nanofertilizer. In our study, we used irregular-shaped ZnO NPs, but it would be interesting in future studies to employ different-shaped ZnO NPs (e.g., rod or spherical), since the size and the shape of NPs can influence their impact on photosynthetic function and thus their utility in agriculture.

## Figures and Tables

**Figure 1 materials-16-05846-f001:**
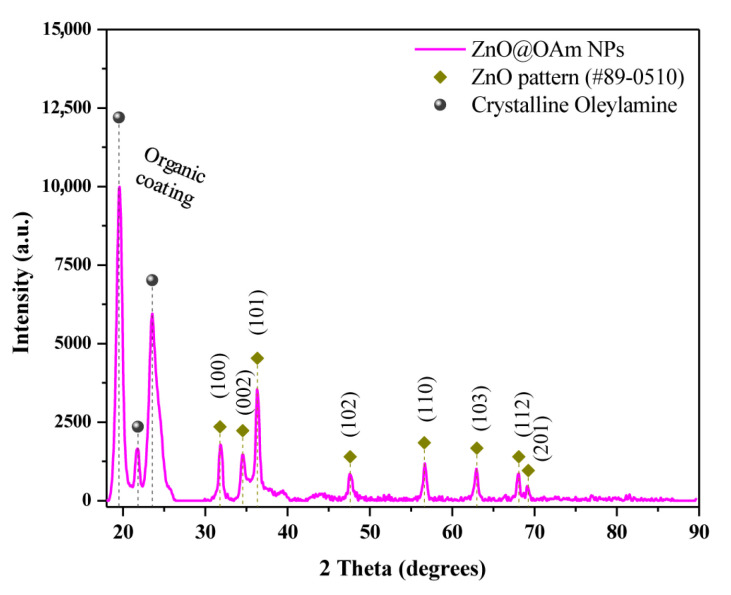
X-ray diffraction spectrum of ZnO@OAm NPs with the corresponding diffraction peaks of zinc oxide.

**Figure 2 materials-16-05846-f002:**
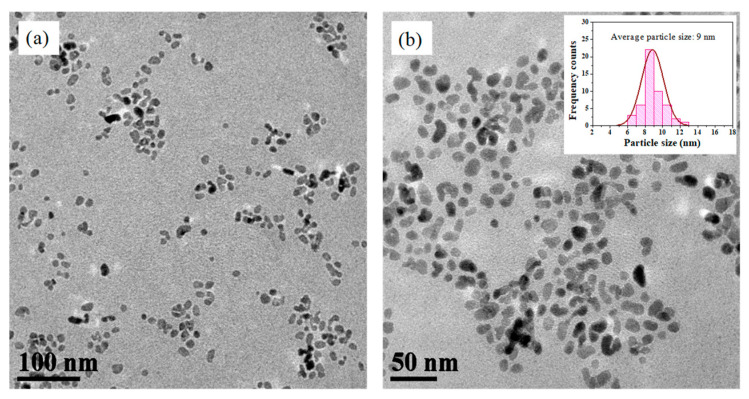
TEM images in scale 100 nm (**a**); and scale 50 nm (**b**) with a size distribution histogram and Gaussian fitting curve (insert) for average particle size 9 ± 1.25 nm (mean size ± SD).

**Figure 3 materials-16-05846-f003:**
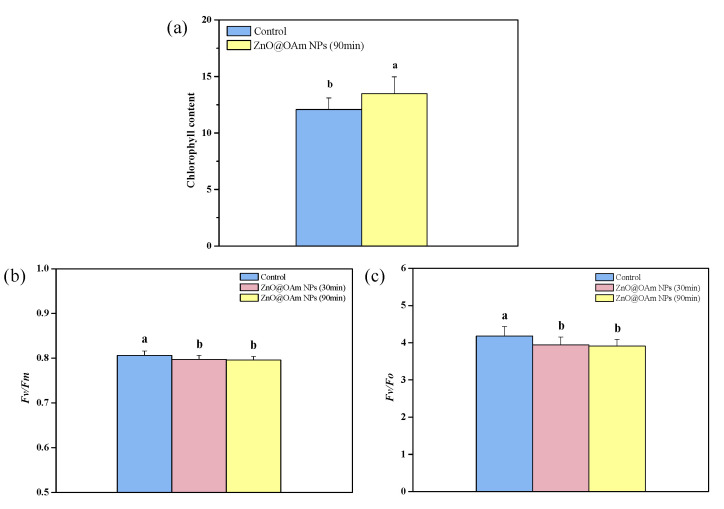
Chlorophyll content (**a**); *Fv*/*Fm* (**b**); and efficiency of the OEC (*Fv*/*Fo*) (**c**), of tomato leaflets 30 and 90 min after the spraying of tomato plants with distilled water (control) or with 15 mg L^−1^ ZnO@OAm NPs. Columns having different lowercase letters are statistically different (*p* < 0.05). Bars in columns represent standard deviation.

**Figure 4 materials-16-05846-f004:**
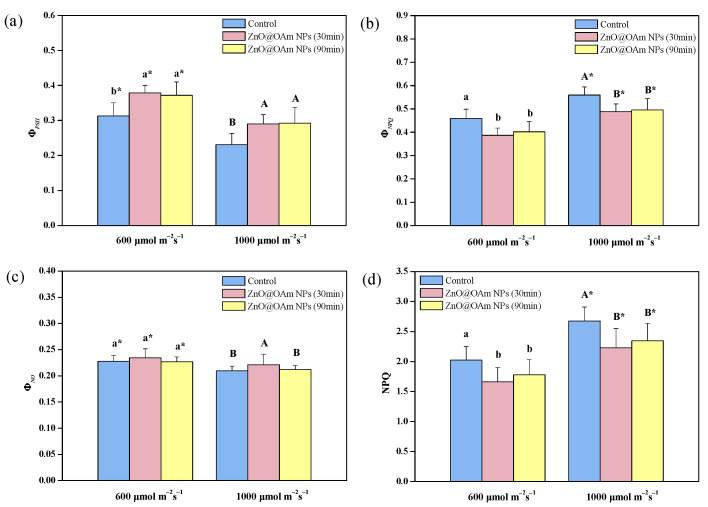
The partitioning of the absorbed light energy. The Φ*_PSII_* (**a**); Φ*_NPQ_* (**b**); Φ*_NO_* (**c**); and the heat dissipation of excitation energy, NPQ (**d**); of tomato leaflets 30 and 90 min after tomato plants were sprayed with distilled water (control) or with 15 mg L^−1^ ZnO@OAm NPs. Bars in columns represent standard deviation. Different lowercase letters show statistically differences at the growth light (GL, 600 μmol photons m^−2^ s^−1^) intensity, while capital letters show statistically differences at the high light (HL, 1000 μmol photons m^−2^ s^−1^) intensity (*p* < 0.05). Significant differences (*p* < 0.05) between GL and HL for the same treatment are indicated by an asterisk (*).

**Figure 5 materials-16-05846-f005:**
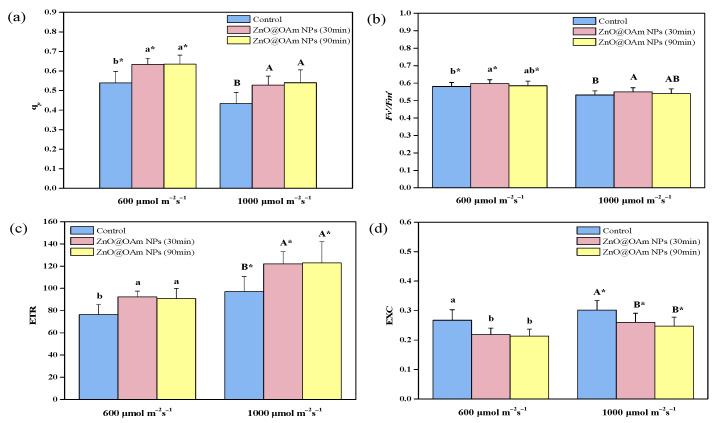
The fraction of open PSII reaction centers (q*p*) (**a**); the efficiency of the open PSII centers (F*v*’/F*m*’) (**b**); the electron transport rate (ETR) (**c**); and the excess excitation energy (EXC) (**d**); of tomato leaflets 30 and 90 min after tomato plants were sprayed with distilled water (control) or with 15 mg L^−1^ ZnO@OAm NPs. Bars in columns represent standard deviation. Different lowercase letters show statistical differences at the growth light (GL, 600 μmol photons m^−2^ s^−1^) intensity, while capital letters show statistical differences at the high light (HL, 1000 μmol photons m^−2^ s^−1^) intensity (*p* < 0.05). Significant differences (*p* < 0.05) between GL and HL for the same treatment are indicated by an asterisk (*).

**Figure 6 materials-16-05846-f006:**
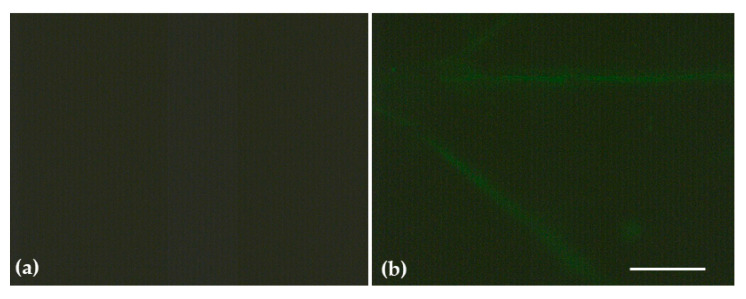
Hydrogen peroxide (H_2_O_2_) generation on tomato leaflets 30 min after tomato plants were sprayed with distilled water (control) (**a**); or with 15 mg L^−1^ ZnO@OAm NPs (**b**). Hydrogen peroxide generation is visible by a very light green color on tomato leaflets after being sprayed with 15 mg L^−1^ ZnO@OAm NPs (**b**). Scale bar: 500 μm.

## Data Availability

The data presented in this study are available in this article.

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
