# Peer review of "Impact of Coated Zinc Oxide Nanoparticles on Photosystem II of Tomato Plants"

_materials, 2023, doi:10.3390/ma16175846_

Round 1

Reviewer 1 Report

In the manuscript titled “Impact of Coated Zinc Oxide Nanoparticles on Photosystem II 2 of Tomato Plants,” the authors have reported the effect of ZnO@OAm NPs on tomato chlorophyll content, reactive oxygen species (ROS) generation, and photosynthetic function. While these NPs increase chlorophyll content but malfunction the oxygen-evolving complex (OEC) of photosystem II (PSII). Due to the decreased photoprotective mechanism of non-photochemical quenching (NPQ) and to the donor-side photoinhibition that is associated with the ROS formation, there is a slight increase in ROS accumulation in tomato leaf veins treated with ZnO@OAm NPs. The larger light-harvesting complexes (LHCs), due to the increased chlorophyll content, resulted to a higher amount of light energy capture that produced an increased electron transport rate (ETR) 29 and quantum yield of PSII photochemistry (ΦPSII). In addition, ZnO@OAm NPs decreased excess excitation energy at PSII, indicating improved PSII efficiency. The authors present the experimental details and analysed data; however, I think the manuscript has the following significant issues.

1.     English looks wobbly at multiple places. For example, line 49 ‘to identify the toxicity caused be nanoparticles’ should be ‘caused by’. Line 101 ‘Briefly, Zn(acac)2 (1.06 mmol) was mixed and dissolved in 4 mL of OAm, under stirring at 30 °C for 15 min.’ The sentence can be rewritten as ‘Briefly, Zn(acac)2 (1.06 mmol) was dissolved in 4 mL of OAm, and mixed well under stirring at 30 °C for 15 min.’

2.     Lines 119-122, The sentence ‘The optical properties of NPs in water-ethanol (3:1) solution and their band gap value were determined by using a UV-Vis spectrophotometer (V-750, 120 Jasco) and based on Tauc’s formula (αhυ)1/n = C (hυ − Eg) [31,32] were dispersed in an ethanol–water mixture with a ratio of 1:3 at a concentration of 15 mg L−1’ shows two different ratios of water-ethanol for collecting UV and final dispersion. Is this typo? If not, how do the ratio affect stability?

3.     Chlorophyll usually shows absorption maxima for either the Soret band 350-450 nm region or Qy 670-690 nm region. Could you please explain the absorption band corresponding to 620 and 920 nm written in line 143?

4.     Did the fluorescence spectrum shows same intensity and peaks after sprayings ZnO@OAm NPs?

5.     Rather than saying PSII, could it be written that energy level matches with the Soret band of various pigments present at PSII?

6.     Why the particle size is so large in DLS than in the TEM images? Does that mean 15 min sonication before DLS measurements didn't help to prevent aggregation?

7.     From Figure 4 (a), the increase looks somewhere at 10 % at growth light and high light spraying with Zno@OAm NPs. The authors should clarify what is the relative increase.

8.     In photosynthetic systems, carotenoids play an important role in NPQ and reducing singlet oxygen. What is the effect of ZnO@OAm NPs on carotenoids? Is it possible that ZNO@OAm is indirectly involved in NPQ with enhancing carotenoids?

9.     Does ETR increase in close PSII or open? Is it possible that the ZNO@OAm affects the local protein environment which facilitates ETR?

After addressing the issues mentioned above, I recommend the manuscript is publishable in a revised manner.

English looks wobbly in multiple places. 

Author Response

In the manuscript titled “Impact of Coated Zinc Oxide Nanoparticles on Photosystem II 2 of Tomato Plants,” the authors have reported the effect of ZnO@OAm NPs on tomato chlorophyll content, reactive oxygen species (ROS) generation, and photosynthetic function. While these NPs increase chlorophyll content but malfunction the oxygen-evolving complex (OEC) of photosystem II (PSII). Due to the decreased photoprotective mechanism of non-photochemical quenching (NPQ) and to the donor-side photoinhibition that is associated with the ROS formation, there is a slight increase in ROS accumulation in tomato leaf veins treated with ZnO@OAm NPs. The larger light-harvesting complexes (LHCs), due to the increased chlorophyll content, resulted to a higher amount of light energy capture that produced an increased electron transport rate (ETR) 29 and quantum yield of PSII photochemistry (ΦPSII). In addition, ZnO@OAm NPs decreased excess excitation energy at PSII, indicating improved PSII efficiency. The authors present the experimental details and analysed data; however, I think the manuscript has the following significant issues.

  1. English looks wobbly at multiple places. For example, line 49 ‘to identify the toxicity caused be nanoparticles’ should be ‘caused by’. Line 101 ‘Briefly, Zn(acac)2 (1.06 mmol) was mixed and dissolved in 4 mL of OAm, under stirring at 30 °C for 15 min.’ The sentence can be rewritten as ‘Briefly, Zn(acac)2 (1.06 mmol) was dissolved in 4 mL of OAm, and mixed well under stirring at 30 °C for 15 min.’

Our response: Line 49 as well as line 109 were corrected accordingly.

“to detect the toxicity caused by nanoparticles”

 “Briefly, Zn(acac)2 (1.06 mmol) was dissolved in 4 mL of OAm, and mixed well under stirring at 30 °C for 15 min.”

  1. Lines 119-122, The sentence ‘The optical properties of NPs in water-ethanol (3:1) solution and their band gap value were determined by using a UV-Vis spectrophotometer (V-750, 120 Jasco) and based on Tauc’s formula (αhυ)1/n = C (hυ − Eg) [31,32] were dispersed in an ethanol–water mixture with a ratio of 1:3 at a concentration of 15 mg L−1’ shows two different ratios of water-ethanol for collecting UV and final dispersion. Is this typo? If not, how do the ratio affect stability?

Our response: We thank the reviewer for the comment. It was typo, the same ratio of solvents has been used. The nanoparticles were dispersed in an ethanol/water solvent mixture at a ratio of 1:3 with a final nanoparticle concentration of 15 mg/L. The sentence was corrected in Lines 127-129 and referred: “The optical properties of NPs in ethanol/water (1:3) solution and their band gap value were determined (15 mg L−1) by using a UV-Vis spectrophotometer (V-750, Jasco) and based on Tauc’s formula [31,32].”

  1. Chlorophyll usually shows absorption maxima for either the Soret band 350-450 nm region or Qy 670-690 nm region. Could you please explain the absorption band corresponding to 620 and 920 nm written in line 143?

Our response: We corrected the sentence by coping from the instruments’ web page http://www.hansatech-instruments.com/product/cl-01-chlorophyll-content-meter/

“using dual-wavelength optical absorbance (660 nm and 940 nm wavelength)”

  1. Did the fluorescence spectrum shows same intensity and peaks after sprayings ZnO@OAm NPs?

Our response: With the Imaging PAM Fluorometer M-Series MINI-Version (Heinz Walz GmbH, Effeltrich, Germany) we do not obtain fluorescence spectrum.

  1. Rather than saying PSII, could it be written that energy level matches with the Soret band of various pigments present at PSII?

Our response: We used the nomenclature of “Kramer, D.M.; Johnson, G.; Kiirats, O.; Edwards, G.E. New fluorescence parameters for the determination of QA redox state and excitation energy fluxes. Photosynth. Res. 2004, 79, 209–218.”

  1. Why the particle size is so large in DLS than in the TEM images? Does that mean 15 min sonication before DLS measurements didn't help to prevent aggregation?

Our response: The sizes are usually different since DLS refers to suspension, which can include aggregates (Brownian motion) and detects a size distribution based on intensity, which may emphasize larger particles or aggregates, while TEM provides direct visualization of dry particles or a minor level of agglomeration.

Despite the 15-minute sonication prior to DLS measurements, it is possible that some degree of aggregation still occurred. Sonication assists in dispersing particles, but the efficiency can vary based on factors like particle concentration, solvent medium, and the power and frequency of sonication.

  1. From Figure 4 (a), the increase looks somewhere at 10 % at growth light and high light spraying with ZnO@OAm NPs. The authors should clarify what is the relative increase.

Our response: We give the percentage increases for Figure 4 (a) on lines 276-279. “The effective quantum yield of PSII photochemistry (ΦPSII), increased (p<0.05) by 21% and 19% after 30- and 90-min spray with ZnO@OAm NPs at growth light (GL, 600 μmol photons m−2 s−1), and by 26% and 27% at high light (HL, 1000 μmol photons m−2 s−1) respectively, compared to controls (Figure 4a).”

  1. In photosynthetic systems, carotenoids play an important role in NPQ and reducing singlet oxygen. What is the effect of ZnO@OAm NPs on carotenoids? Is it possible that ZNO@OAm is indirectly involved in NPQ with enhancing carotenoids?

Our response: Since NPQ decreased after spraying, it seems that there was not any carotenoid enhancement.

  1. Does ETR increase in close PSII or open? Is it possible that the ZnO@OAm affects the local protein environment which facilitates ETR?

Our response: The quantum yield of PSII photochemistry (ΦPSII) and thus ETR (=ΦPSII × PAR × c × abs) increased due to the increased fraction of open PSII reaction centers (Figure 5a)

Reviewer 2 Report

. The authors have addressed this issue in a very interesting way.

The authors have presented the results of their work in a very interesting and clear way. All the results were supported by the experiments carried out. The results are presented in figures. Thus, the work is very promising and can be recommended for publication in present form.

Author Response

The authors have addressed this issue in a very interesting way.

The authors have presented the results of their work in a very interesting and clear way. All the results were supported by the experiments carried out. The results are presented in figures. Thus, the work is very promising and can be recommended for publication in present form.

This article is very interesting from a scientific as well as agricultural point of view. Mainly when it comes to growing tomatoes. The authors have addressed this issue in a very interesting way.

The authors have presented the results of their work in a very interesting and clear way. All the results were supported by the experiments carried out. The results are presented in figures. Thus, the work is very promising and can be recommended for publication in present form.

Comments

The only one comment. I would correct the font size (scale bars) on the drawings. They will be more readable.

Our response: We will have in mind your comment in our next manuscript to increase the font size on the drawings.

Reviewer 3 Report

Please find the attached review file.

Author Response

The manuscript titled “Impact of Coated Zinc Oxide Nanoparticles on Photosystem II of Tomato Plants” by Tryfon et al. provides a comprehensive investigation into the effects of oleylamine-coated ZnO nanoparticles on photosynthetic processes in tomato plants. The study offers valuable insights into the interaction between nanoparticles and plants, with potential implications for crop yield enhancement. The authors explore the influence of nanoparticle characteristics on photosynthetic efficiency, reactive oxygen species generation, and chlorophyll content. While the study contributes important findings to the field, there are several general aspects that should be carefully addressed and considered before the manuscript's consideration and publication:

  1. Abstract should be improved and rewritten taken into account these points:
  2. Briefly describe the methodology for obtaining ZnO@OAm NPs, the main objective of developing this product, and conclude with the positive and negative implications accordingly.

Our response: Abstract has been reformed while the positive implications of solvothermally prepared NPs has been mostly emphasized in Introduction due to the limited size (200 words) of the abstract.

  1. Define abbreviations, such as ZnO NPs coated with oleylamine (ZnO@OAm NPs), at their first mention in the manuscript to enhance clarity for readers.

Our response: All abbreviations are defined upon their initial appearance in the manuscript.

  1. Simplify sentence structures for better readability, emphasizing key findings for clearer communication.

Our response: We addressed your comment in the revised manuscript trying to simplify sentence structures and emphasizing key findings for clearer communication.

  1. While acknowledging the potential benefits of ZnO NPs, emphasize the practical implications of observed concerns like photoinhibition and oxidative stress.

Our response: It would be a speculation if we will try to emphasize the practical implications of observed photoinhibition and oxidative stress.

  1. Briefly introduce the agricultural context to underscore the research's relevance and innovation in addressing contemporary agricultural needs.

Our response: Since there is a word limit to about 200 words for the Abstract, we include this information on Introduction.

  1. Additionally, the authors should highlight quantitative results to provide more precise and concrete evidence supporting their conclusions.

Our response: We included such data in the revised Abstract.

  1. Introduction:
  2. The authors are encouraged to emphasize the significance of zinc metal oxide nanoparticles in a more comprehensive manner. Additionally, highlighting their distinctive attributes across various applications, such as the examples mentioned in reference to "doi.org/10.1039/D3NJ00131H," would further strengthen the credibility of this study.

Our response: Introduction enriched with the suggested applications and the mentioned reference has been added. Additionally, the positive implications of solvothermally prepared NPs that are already referred in Lines 83-84, was enlarged in Lines 103-107: "The solvothermal approach is a versatile technique for crafting a diverse range of materials, from metals and ceramics to semiconductors and polymers. This procedure employs solvents under varying pressures and temperatures, promoting precursor interactions offering good homogeneity insize and morphology. Meanwhile, method's adaptability, cost-effectiveness, and simplicity are its core strengths [31]."

Also, in Introduction was added "Thus, zinc oxide nanoparticles (ZnO NPs) have gained significant attention due to their unique properties, including high electron mobility, wide bandgap, and exceptional photocatalytic activity [19,20] while is also widely used in several medical based applications as cosmetics, drug delivery agents and antibacterial activities against both Gram-positive and Gram-negative pathogenic strains [e.g. 21]."

[21] Abdullah, J.A.A.; Jiménez Rosado, M.; Guerrero, A.; Romero, A. Eco-friendly synthesis of ZnO-nanoparticles using Phoenix dactylifera L., polyphenols: physicochemical, microstructural, and functional assessment. New J. Chem. 2023, 47, 4409–4417. doi.org/10.1039/D3NJ00131H

  1. The authors should provide a concise overview of the diverse methods employed for acquiring ZnO-NPs, outlining both their benefits and limitations. Referring to the aforementioned source could enhance this discourse along with other relevant references.

Our response: It is out of the scope of the present study to discuss on the diversity of the preparation methods. To our opinion this is covered by review papers.

  1. The authors should conclude the introduction by highlighting the main novelty of the study.

Our response: We added such sentence in lines 90-92.

  1. Materials and methods
  2. The materials, including DCF-DA and other substances, should be first mentioned in the materials section.

Our response: In the first section we presented the chemicals and reagents that were used for the synthesis of ZnO@OAm NPs and all other materials are mentioned on the method that they are used.

  1. Include more detailed information in section 2.1. Solvothermal Synthesis of ZnO@OAm NPs. Additionally, ensure that this section is supported by appropriate references to validate the chosen experimental parameters and methods.

Our response: In response to this suggestion, section 2.1 has been expanded to provide a more comprehensive description of the synthesis process. Additionally, to validate the chosen experimental parameters and methods, this section has been bolstered with pertinent references, including the statement: " The synthesis of ZnO@OAm NPs was performed according to previous study [32] with certain modifications."

  1. However, it's essential to acknowledge the potential impact of washing solvents and solvothermal treatment temperatures on the synthesized nanoparticles, which could lead to either partial or complete alterations. Therefore, the authors are strongly urged to explore the potential effects of these parameters and incorporate them into their analysis. To substantiate their findings, it's advisable for the authors to incorporate pertinent information from sources such as "doi:10.3390/nano13152242" and "doi.org/10.3390/ma16051798.

Our response: Herein, the intrinsic properties of ZnO@OAm NPs was explored on PSII photochemistry in tomato plants. The above references are not straight forward to the present study as they are referred to other material (iron oxide nanoparticles) and the particularity is given. However, they are interesting refs and we’ll keep in mind in future studies to be included. Also, at Results and Discussion in XRD analysis crystallinity degree was added based on Abdullah et al., 2023:

"The ZnO@OAm NPs exhibited approximately 91% crystallinity, a value that appears to be influenced by the solvothermal approach and/or the ethanol that was employed as a solvent during the repeating washing steps [44]."

[44] Abdullah, J.A.; Díaz-García, Á.; Law, J.Y.; Romero, A.; Franco, V.; Guerrero, A. Quantifying the structure and properties of nanomagnetic iron oxide particles for enhanced functionality through chemical synthesis. Nanomaterials 2023, 13, 2242. https://doi.org/10.3390/nano13152242.

  1. Authors should elaborate on their experimental methodology, including details on the apparatus used and the information gathered from each technique. Referring to previous sources can enhance this section and justify the chosen methods, ensuring scientific rigor and supporting the study's results and conclusions.

Our response: The session of Materials and Methods includes comprehensive information about the equipment, techniques, solvents, and equations employed during the study. Additionally, to validate and justify our chosen methods, pertinent references have been cited within this section. We believe that this approach ensures scientific rigor and provides strong support for the results and conclusions of our study.

  1. Results.
  2. It would be valuable to know whether the authors also characterized the uncoated ZnO-NPs. This additional step could provide insightful information, especially in demonstrating the effect of the OAm coating.

Our response: Uncoated ZnO-NPs have not been prepared; the selection of the coating agent can profoundly affect multiple parameters, particularly the size and morphology of the resultant nanoparticles, as it is demonstrated before in previous studies by us and others (Tryfon et al., 2019; Giannousi et al., 2022; Tryfon et al., 2023).

  1. Figure 1 depicts distinct crystalline patterns in ZnO-NPs and OAm. Consequently, it is recommended that the authors furnish details about the size of the coated peaks and elucidate its impact on crystalline size – whether it has increased or decreased

– while also offering the crystallinity index. The preceding article could provide valuable insights to support this discussion.

Our response: We calculated the crystalline grain size for the peaks at lower degrees, which are associated with the presence of crystalline oleylamine. Scherrer's equation provided a size of 11-12 nm in that case, which is lower than the 19 nm mean value for the crystalline grain size of the ZnO. However, we believe that such comparison is not so relevant and not that helpful, because in that case we attempt to compare different entities, and it is not easy to deduce a meaningful insight out of this.

Nevertheless, as the reviewer suggested, we calculated the crystallinity degree for our sample, yielding a value of approximately 91.2%. Such crystallinity degree suggests that a small portion of the product corresponds to some non-crystalline residual oleylamine and/or to possible amorphous domains being present in the ZnO NPs.

"The ZnO@OAm NPs exhibited approximately 91% crystallinity, a value that appears to be influenced by the solvothermal approach and/or the ethanol that was employed as a solvent during the repeating washing steps [44]."

  1. To enhance visualization and presentation, the authors could consider incorporating supplementary figures (Fig. S1-S4) into the results section, specifically within a categorized group (a, b, c, and d) under the characterization section. This would consolidate all figures, making it easier to identify and comprehend the results.

Our response: We thank the reviewer for thoughtful suggestion on consolidating the figures for enhanced visualization. However, we deliberately chose to present the techniques separately for the solid and liquid phases as they provide distinct information. Combining them might lead to complexities in interpreting the data. Furthermore, six figures already included in the main manuscript, adding more might make it cumbersome for readers. We believe this structure maintains clarity while offering comprehensive data.

  1. Conclusion
  2. The conclusion effectively summarizes the study's findings and outlines future perspectives. However, it could be enhanced by incorporating a discussion of the study's limitations. This addition would provide a more comprehensive overview of the research and its potential implications.

Our response: We added in the discussion possible limitations for the application of ZnO NPs in agriculture that depend on the differential shape of ZnO NPs (e.g., rod or spherical), that could have differential impact on photosynthetic function.